# The KU-ISPL entry to the GENEA Challenge 2023-A Diffusion Model for Co-speech Gesture generation

Gwantae Kim
kgt1103211@korea.ac.kr
Korea University
Seoul, South Korea

Yuanming Li
lym7499500@korea.ac.kr
Korea University
Seoul, South Korea

Hanseok Ko
hsko@korea.ac.kr
Korea University
Seoul, South Korea

## ABSTRACT

This paper describes a diffusion model for co-speech gesture generation presented by KU-ISPL entry of the GENEA Challenge 2023. We formulate the gesture generation problem as a co-speech gesture generation problem and a semantic gesture generation problem, and we focus on solving the co-speech gesture generation problem by denoising diffusion probabilistic model with text, audio, and pre-pose conditions. We use the U-Net with cross-attention architecture as a denoising model, and we propose a gesture autoencoder as a mapping function from the gesture domain to the latent domain. The collective evaluation released by GENEA Challenge 2023 shows that our model successfully generates co-speech gestures. Our system receives a mean human-likeness score of 32.0, a preference-matched score of appropriateness for the main agent speech of 53.6%, and an interlocutor speech appropriateness score of 53.5%. We also conduct an ablation study to measure the effects of the pre-pose. By the results, our system contributes to the co-speech gesture generation for natural interaction.

## CCS CONCEPTS

• **Computing methodologies → Animation**; • **Human-centered computing → Human computer interaction (HCI)**.

## KEYWORDS

GENEA Challenge, co-speech gesture generation, diffusion, neural networks, generative models

**ACM Reference Format:**
Gwantae Kim, Yuanming Li, and Hanseok Ko. 2023. The KU-ISPL entry to the GENEA Challenge 2023-A Diffusion Model for Co-speech Gesture generation. In *Proceedings of 25th ACM International Conference on Multimodal Interaction (ICMI'23)*. ACM, New York, NY, USA, 8 pages. https://doi.org/XXXXXXX.XXXXXXX

## 1 INTRODUCTION

Synthesizing synchronized and human-like gestures performs crucial roles to improve immersion, engagement, and naturalness for embodied virtual agents and humanoid robots. During the human-computer interaction(HCI) process, human uses both verbal and non-verbal expressions to provide their intent to the interlocutor. Gesture generation, which is one of the main challenges for non-verbal interaction, aims to synthesize natural-looking and meaningful human gestures. The task can be separated whether verbal expression exists or not. When verbal expressions, such as audio or text, are given, the gesture generation model focuses on making gestures that emphasize the meaning of verbal expressions. In the other case, the model should generate gestures that deliver the intent whether verbal expressions are given or not. We define the task with verbal information as co-speech gesture generation and the task that focuses on synthesizing meaningful body motions that deliver intent as semantic gesture generation. In this research, we focus on generating high-fidelity co-speech gestures.

There are many challenges for the co-speech gesture generation. The first is timing synchronization. Since the speech and gestures are shown to the interlocutor sequentially, he or she will be confused if gestures depart from speech. For example, if the start and end timing of the gestures slightly differs from speech, the users will think that it is an implemental error. A more detrimental situation is traffic jams during continuous generation. Once the timing is out of sync, the timing between speech and gestures is continually departed and the discomfort will be gradually increased. With similar thinking, semantic synchronization, which is the second challenge, is also important to deliver proper intent. For example, when people say "I disagree." by nodding, the interlocutor will be confused that it is positive or negative.

The third obstacle is noise robustness. 3D pose estimation or motion capture is utilized to acquire gesture data. However, the quality of raw data obtained by 3D pose estimation is not enough because the algorithm is basically image-to-3D reconstruction, which is a one-to-many problem. The motion capture is better, but it is too expensive and time-consuming. To secure quality, the cost is increased exponentially. Therefore, the raw data may contain noise. Since training with noisy data hurts both quantitative and qualitative performance, a workaround such as pre-processing or noise-robust training is needed.

To tackle these problems, deep learning-based approaches have been applied to generating co-speech gestures, recently. There are three types of training strategies: reconstruction-based method[15, 18, 34], generative adversarial network(GAN)[8] based method[25, 33], and diffusion[7, 12] based methods[3, 5, 38]. The reconstruction-based co-speech gesture generation methods directly estimate gestures from text or audio. Although the methods induce reasonable results in terms of joint error, disadvantages are seen in terms of diversity. To generate various results without quantitative performance degradation, GAN-based co-speech gesture generation models are trained by controlling the weight between reconstruction

loss and adversarial loss. Recently, denoising diffusion probabilistic models(DDPMs) are achieving huge success in the generative model and computer vision fields and expanding to other research fields[14, 24]. Especially, the diffusion model could synthesize various images that reflect input conditions, even if its semantic space is large. Since the semantic space of the speech for co-speech gesture generation is large, the diffusion model may help to synthesize various and synchronized results. Therefore, the goal of the paper is to find a suitable diffusion model structure for co-speech gesture generation.

In this paper, we propose a diffusion-based co-speech gesture generation method. We establish a gesture autoencoder to project from gesture space to feature space and vice versa. The model was configured to select suitable features according to the characteristics of the gesture data. We also present how to deliver audio and text information to the diffusion model. We use validated audio features and the pre-trained language model to provide rich features.

The data and evaluations are provided by GENEA Challenge 2023[20]. Thanks to the good-quality data, the noise robustness problem is under control and we can focus on the synchronizing problems. The evaluations, which contain human likeness, and appropriateness for the main agent and interlocutor, are also well-formulated to measure the generation performance. The code is available here[1].

## 2 RELATED WORKS

### 2.1 Co-speech gesture generation

Kucherenko et al. [18] proposed an autoencoder-style audio-to-gesture model with hidden representation learning. The method first find hidden embedding space of gesture by autoencoder and next train the audio encoder to find joint embedding space between audio and gesture. Yoon et al. [34] trained the sequence-to-sequence LSTM model to map text transcriptions to 2D co-speech gestures. Kim et al. [15] trained the transformer-based autoencoder with self-supervised pre-training. These approaches use reconstruction loss to optimize the model. Chang et al.[4] presented a locality constraint attention-based gesture generation model, which is inspired by Tacotron2. StyleGestures [1] uses the method of normalizing flow to generate gestures from speech. Audio2Gestures [22] synthesize gestures using a variational autoencoder. Yoon et al. [33] train the model with adversarial loss and reconstruction loss to generate gestures from trimodal contexts. HA2G [25] adopts a hierarchical decoder to address the structural information of the joint. Gesturemaster[37] used a rhythm embedding module, style embedding module, motion graph construction, and graph-based optimization to extract features and generate gestures.

### 2.2 Semantic gesture generation

Kim et al. [16] generates gestures with the semantics itself or extracted from text. The method with an intent classifier emphasizes co-speech gesture generation. The co-speech gesture model is selected to generate gestures if the intent is unclear, else this method

is used to synthesize gestures. SEEG [23] generates semantic energized co-speech gestures with the semantic prompt gallery, semantic prompter, and semantic energized learning. Gesticulator[19] synchronizes between text and audio features in the encoding phase and generates gestures by autoregression.

### 2.3 Diffusion-based motion generation

Alexanderson et al. [2] proposed conformer[10]-based diffusion models for gesture generation, dance synthesis, and path-driven locomotion. Zhu et al. [38] migrated the diffusion model to speech-driven co-speech gesture generation with diffusion gesture stabilizer and implicit classifier-free guidance. FLAME [17] generates and edits human motion with the pre-trained language model and transformer. Motiondiffuse[36] and MDM[30] also synthesize human motions from text descriptions. [3] learns a gesture-transcript joint embedding space using contrastive learning. The learned embeddings are incorporated into the diffusion model via an adaptive instance normalization layer. [5] synthesize motions by diffusion model using latent space. The motion representations are projected into latent space, diffused, and reconstructed to the original motion space.

## 3 CO-SPEECH GESTURE GENERATION MODEL

Figure 1 depicts an overview of the proposed model to generate high-fidelity co-speech gestures. In this section, we first introduce the problem formulation of co-speech gesture generation (Section 3.1). We propose the gesture autoencoder, which is designed to project gesture space to feature space (Section 3.2). We then present the classifier-free guidance for applying speech conditions to co-speech gestures (Section 3.3). Furthermore, we establish the forward diffusion and the reverse conditional generation process in feature space (Section 3.4).

### 3.1 Problem Formulation

The co-speech gesture training data often consist of 3D pose sequence $\mathbf{x}$, audio $\mathbf{a}$, text(sentence) $\mathbf{s}$, and metadata. The generative model G parameterized by $\theta$ is optimized to synthesize $\mathbf{x}$, which is further conditioned on the audio $\mathbf{a}$, text $\mathbf{s}$, and the pre-defined initial poses $\mathbf{x}_{-1}$ of the M frames. The learning objective of the problem can be formulated as $arg\min_\theta ||\mathbf{x} - G_\theta(\mathbf{a}, \mathbf{s}, \mathbf{x}_{-1})||$.

However, samples in the training data often have a long duration. To reduce the computational cost and memory usage, every modality of the sample is cropped into segments $\mathbf{x} = \{\mathbf{x}_1, ..., \mathbf{x}_i\}$, $\mathbf{a} = \{\mathbf{a}_1, ..., \mathbf{a}_i\}$, and $\mathbf{s} = \{\mathbf{s}_1, ..., \mathbf{s}_i\}$, where $\mathbf{x}_i$ has N frames and $\mathbf{a}_i, \mathbf{s}_i$ have the same time length as $\mathbf{x}_i$. Now the generative model G estimates $\mathbf{x}_i$ from the audio $\mathbf{a}_i$, text $\mathbf{s}_i$, and the M pose frames from previous segment $\mathbf{x}_{i-1}^{(N-M):N}$, instead of synthesizing $\mathbf{x}$ at once. Finally, the generative model G synthesizes the gestures $\{\mathbf{x}_1, ..., \mathbf{x}_i\}$ continuously.

The model is autoregressive because the poses generated by the previous segment are used to synthesize the current segment, and stochastic because the initial diffusion feature map is random noise.

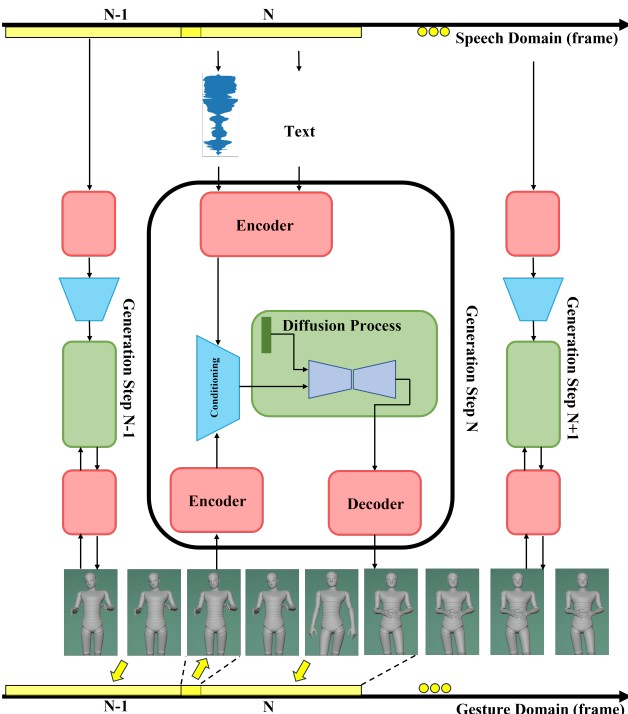

**Figure 1: Overview of the proposed diffusion-based co-speech gesture generation method. The model is autoregressive and probabilistic. For the N-th generation, audio, text, and pre-poses are projected to the latent space and used to conditions. The initialized Gaussian noise is iteratively diffused by the reverse process. The output latent vector is reconstructed to the gesture space by the decoder.**

## 3.2 Gesture Autoencoder

In the Stable Diffusion[28], the latent diffusion model provides flexible, computationally tractable, and sometimes achieving quality improvement. The gesture autoencoder focus on finding good latent embedding space projected from gesture space. The gesture autoencoder consists of two autoencoder models: pose autoencoder and motion autoencoder. Since the gesture is the sequential pose data, we design the pose autoencoder for projecting the raw pose space to latent space, and the motion autoencoder to find correlations along the time axis.

The pose encoder and decoder consist of 3 fully-connected layers with dropout[29] and GELU activation function[11] each. The input poses sequence $\mathbf{x}^{N \times 3J}$ is projected to $\mathbf{z'}^{N \times D}$ by the pose encoder, where $\mathbf{z'}$ denotes mid-level hidden representation, J is the number of joints, and D is the dimension of $\mathbf{z'}$, and the pose decoder performs reverse projection. The pose autoencoder is first trained with L1 reconstruction loss. Once the pose autoencoder is optimized, the parameters are frozen in the rest training stages such as diffusion training stage.

The motion autoencoder aims to capture sequential information of the data. Thus, the motion encoder and decoder consist of 3 gated recurrent units(GRU) layers[6] and 3 multi-head self-attention

layers[31], which have strong capacity in sequential data modeling. The motion encoder is formulated

$$\mathbf{z} = MHSA(GRU(\mathbf{z'}))\tag{1}$$

where $MHSA(X) = Attention(X, X, X)$. The attention mechanism is

$$Attention(Q, K, V) = softmax(\frac{QK^T}{\sqrt{d}}) \cdot V\tag{2}$$

where Q, K, and V are the query, key, and value from the feature matrix, d is the channel dimension, and T is the matrix transpose operation.

The mid-level hidden representation $\mathbf{z'}^{N \times D}$ is projected to $\mathbf{z}^{N \times D}$ by the motion encoder, where $\mathbf{z}$ denotes hidden representation in feature space, and the motion decoder performs reverse projection. The motion autoencoder is individually trained with L1 reconstruction loss. The parameters of the motion autoencoder are also frozen after this training stage.

## 3.3 Conditioning

The diffusion models are theoretically capable of modeling the conditional distribution $p(z|y)$. This can be implemented with a conditional denoising autoencoder $\epsilon_\theta(z_t, t, y)$, where $y \in \{\mathbf{a}, \mathbf{s}, \mathbf{z_{i-1}}\}$, to address the generation process through inputs y. To combine conditional information and latent vector in the U-Net backbone, we use a cross-attention mechanism, which is used in Stable Diffusion[28].

The three modalities, which are audio, text, and pre-pose, are used as conditions in the diffusion process. The pre-processed audio features, text features, and pre-pose features are projected to the embedding vectors by fully-connected layers. These three embedding vectors are added to the time embedding vector and propagate the information of each modality to the denoising U-Net model.

## 3.4 Diffusion

DDPMs define the latent variable models of the form $p_\theta(x_0) = \int p_\theta(x_{0:T}) dx_{1:T}$, where $x_{1:T}$ are latent variables in the same sample space as $x_0$ with the same dimensionality.

The forward process, which is also called the diffusion process, approximates the posterior distribution $q(x_{1:T}|x_0)$ by the Markov chain that gradually adds Gaussian noise to the data according to the variance schedule $\beta_1, ..., \beta_T$:

$$q(x_{1:T}|x_0) = \prod_{t=1}^{T} q(x_t|x_{t-1}),\tag{3}$$

where

$$q(x_t|x_{t-1}) = \mathcal{N}(x_t; \sqrt{1-\beta_t}x_{t-1}, \beta_t \mathbf{I}).\tag{4}$$

The forward process variances $\beta_t$ can be learned by reparameterization or held constant as hyperparameters. Since our model uses gesture autoencoder for mapping from pose to latent embeddings, the latent embeddings are gradually corrupted by noise, which finally leads to a pure white noise when T goes to infinity. Therefore, the prior latent distribution of $p(x_T)$ is $\mathcal{N}(x_t; \mathbf{0}, \mathbf{I})$ with only information of Gaussian noise.

The reverse process estimates the joint distribution of $p_\theta(x_{0:T})$. It is defined as a Markov chain with learned Gaussian transitions

starting at $\mathcal{N}(x_t; \mathbf{0}, \mathbf{I})$:

$$p_\theta(x_{0:T} = p(x_T) \prod_{t=1}^{T} p_\theta(x_{t-1}|x_t), \quad (5)$$

where

$$p_\theta(x_{t-1}|x_t) = \mathcal{N}(x_{t-1}; \mu_\theta(x_t, t), \Sigma_\theta(x_t, t)). \quad (6)$$

The corrupted noisy latent embedding $x_t$ is sampled by $q(x_t|x_0) = \mathcal{N}(x_t; \sqrt{\bar{\alpha}_t}x_0, (1 - \bar{\alpha}_t)\mathbf{I})$, where $\alpha_t = 1 - \beta_t$ and $\bar{\alpha}_t = \prod_{s=1}^{t} \alpha_s$.

Since the problem is co-speech gesture generation, which is a conditional generation problem, we have to provide additional inputs $\mathbf{a}, \mathbf{s}$, and $\mathbf{z_{i-1}}$ to the model. Therefore, these conditions are injected into the generation process. The reverse process of each timestep can be updated for our problem as:

$$p_\theta(z_{t-1}|z_t, y) = \mathcal{N}(x_{t-1}; \mu_\theta(z_t, t, y), \beta_t \mathbf{I}). \quad (7)$$

The reverse process is started by sampling a Gaussian noise $z_t \mathcal{N}(\mathbf{0}, \mathbf{I})$ and following the Markov chain to iteratively denoise the latent variable $x_t$ via Eq. 7 to get the original latent vector $z_0$.

The variational lower bound on negative log-likelihood is used to optimize the diffusion model. We follow [12] to simplify the training objective to the ensemble of MSE losses as:

$$L(\theta) = \mathbb{E}_{t,x_0,\epsilon} [||\epsilon - \epsilon_\theta(\sqrt{\bar{\alpha}_t}x_0 + \sqrt{1 - \bar{\alpha}_t}\epsilon, y, t)||^2], \quad (8)$$

where t is uniformly sampled between 1 and T, and $\epsilon$ is initialized as $\mathcal{N}(\mathbf{0}, \mathbf{I})$. The diffusion model is trained by the gradient descent steps on Eq. 8 until converged.

## 4 EXPERIMENT

### 4.1 Data Processing

We trained our model using the GENEA Challenge 2023 dataset[35], derived from the Talking with hands 16.2M dataset[21]. This dataset comprises a training set containing 371 clips, a validation set with 40 clips, and a test set encompassing 70 clips. Each clip consists of audio recordings, transcriptions, gesture motions for the main agent, gesture motions for the interlocutor, and associated metadata. The audio data possesses a sampling rate of 44100Hz. The gesture motions are formatted in BVH (Biovision Hierarchy) format, and their frame rate is set at 30 frames per second (FPS).

Our system exclusively utilizes audio and text data from the main agent, disregarding the interlocutor's information and metadata. We extract the mel-spectrogram, mel-frequency cepstrum coefficients, and prosody features using n-fft=4096 and a hop length of 33ms. To extract audio features, we employed the Librosa[27] package and the Parselmouth[13] library. The network output comprises joint angles relative to a T-pose, with these angles parameterized using the exponential map[9]. Each dimension is normalized to have a mean of zero and a standard deviation of one across the official challenge training set. We selected a total of 26 joints for full-body expression. Subsequently, we apply a Savitzky-Golay filter[26] with a window length of 9 and a polynomial order of 3 to smooth the generated gestures. For text segmentation, we employ a pre-trained text embedding model[32], featuring 1024 dimensions per sentence. We opted for sentence embedding due to its capacity to capture semantic information in contrast to word embeddings. Given that the audio, text, and gesture data are temporally aligned,

**Table 1: Detailed hyperparameters setting**

| Hyperparameter | Value |
|---|---|
| # of joints (J) | 26 |
| # of pre-pose frames (M) | 8 |
| # of frames of the segment (N) | 128 |
| Denoising diffusion steps | 1000 |
| Feature dimension (D) | 128 |
| Condition vector dimension | 512 |
| # of residual blocks per up/downsampling layer | 2 |
| # of up/downsampling layers | 4 |
| # of attention heads | 4 |
| N-FFT | 4096 |
| Hop length [ms] | 33 |
| Text embedding dimension | 1024 |
| optimizer | AdamW |
| learning rate | 1e-4 |
| batch size | 8 |

the timing of audio features, text embeddings, and pose sequences are synchronized.

## 5 DISCUSSION

In this section, we provide some discussions about evaluation results. The submitted co-speech gestures are measured by three aspects: human likeness, appropriateness for agent speech, and appropriateness for the interlocutor. The natural motion, monadic baseline, and dyadic baseline are labeled NA, BM, and BD, respectively. Our submitted entry name is named SA. Our gesture generation system is tested on a Windows 10 desktop with a 3.20GHz i9-12900K CPU, 128GB RAM, and one RTX 3090 GPU.

### 5.1 Human-likeness

The results of the evaluation are presented in Table 2 and Figure 2. Our submitted system achieves a median human-likeness score of 30 and a mean human-likeness score of 32.0. A disparity in human likeness is observed between our entry and natural motions. One of the significant contributing factors to this phenomenon is the lack of structural information. By not capturing the interdependencies among joints, our model generates gestures with a predominant emphasis on arm movements, which tend to exhibit greater motion compared to head or body joints. Since the movement of the center of gravity of the agent is ignored by the above reason, the human likeness score may decrease. Furthermore, our system omits finger motions from its generation process. Another conceivable concern is the effectiveness of smoothing techniques. Despite the application of a smoothing filter, the motions produced by our system sometimes appear to lack smoothness. Potential factors contributing to these results encompass suboptimal optimization of the smoothing filter and an insufficient number of pre-pose instances.

### 5.2 Appropriateness

In respect to the appropriateness of speech exhibited by main agent, Table 3 and Figure 4 provide a description indicating that our entry achieves a preference-matching score of 54.8%. The outcomes of

**Table 2: Summary of the collective perception study with a 0.05 confidence interval about human-likeness. Our entry is SA.**

| Condition | Human-likeness | |
|---|---|---|
| | Median | Mean |
| NA | $71 \in [70, 71]$ | $68.4 \pm 1.0$ |
| SG | $69 \in [67, 70]$ | $65.6 \pm 1.4$ |
| SF | $65 \in [64, 67]$ | $63.6 \pm 1.3$ |
| SJ | $51 \in [50, 53]$ | $51.8 \pm 1.3$ |
| SL | $51 \in [50, 51]$ | $50.6 \pm 1.3$ |
| SE | $50 \in [49, 51]$ | $50.9 \pm 1.3$ |
| SH | $46 \in [44, 49]$ | $45.1 \pm 1.5$ |
| BD | $46 \in [43, 47]$ | $45.3 \pm 1.4$ |
| SD | $45 \in [43, 47]$ | $44.7 \pm 1.3$ |
| BM | $43 \in [42, 45]$ | $42.9 \pm 1.3$ |
| SI | $40 \in [39, 43]$ | $41.4 \pm 1.4$ |
| SK | $37 \in [35, 40]$ | $40.2 \pm 1.5$ |
| SA | $30 \in [29, 31]$ | $32.0 \pm 1.3$ |
| SB | $24 \in [23, 27]$ | $27.4 \pm 1.3$ |
| SC | $9 \in [9, 9]$ | $11.6 \pm 0.9$ |

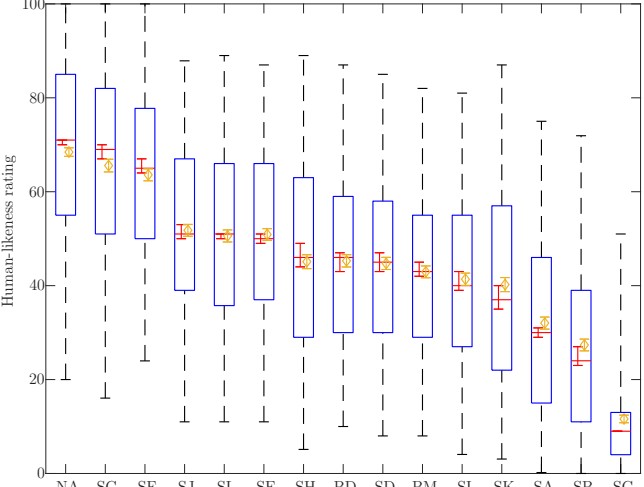

**Figure 2: Box plot visualizing the rating distribution in the human-likeness study. Red bars are the median ratings (each with a 0.05 confidence interval); yellow diamonds are the mean ratings (also with a 0.05 confidence interval). Box edges are at 25 and 75 percentiles, while whiskers cover 95% of all ratings for each condition.**

the assessment, which focuses on the appropriateness of interlocutor speech, are displayed in Table 4 and Figure 6. Our developed system attains a preference-matching score of 53.5%. We present a concise overview of several configurations within our experimental framework, which we posit may contribute to enhancing the appropriateness of gestures about speech. One potential rationale we identify pertains to semantic conditioning. Our system employs

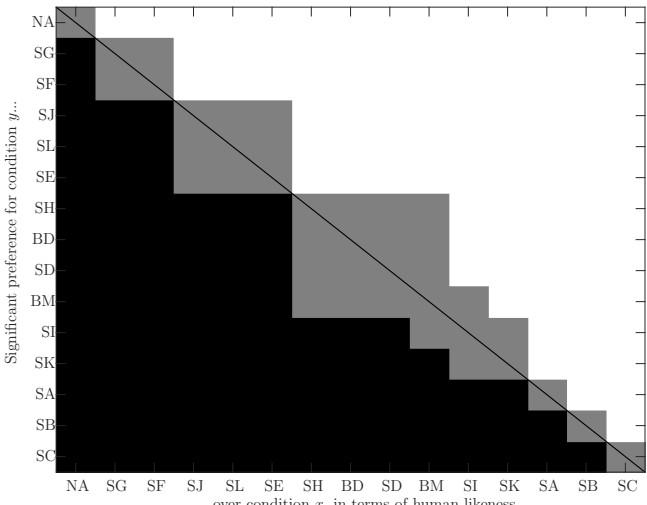

**Figure 3: Significance of pairwise differences between conditions. White means that the condition listed on the $y$-axis rated significantly above the condition on the $x$-axis, black means the opposite ($y$ rated below $x$), and grey means no statistically significant difference at the level $\alpha = 0.05$ after Holm-Bonferroni correction.**

a pre-trained sentence embedding model without fine-tuning. However, numerous textual segments in the data fail to adhere to proper sentence structure. Consequently, the embedding might inaccurately convey the semantics of these text segments. To mitigate this concern, we will change the sentence embedding model to a word embedding model, or utilization of extended segments.

Furthermore, timing synchronization is a consideration. Given that our system incorporates speech features such as mel-spectrogram, MFCC, and prosody to extract temporal information from audio, the model learns to effectively synchronize audio with gestures. Additionally, the pre-pose condition aids in capturing the initiation timing. Consequently, the proposed model demonstrates the capability to regulate the timing of speech onset and pauses.

Moreover, we address the issue of gesture smoothness. The generated gesture results from our system sometimes exhibit irregularity. We hypothesize that the phenomenon may be attributed to the architecture of the pose autoencoder, the pre-poses, and the extent of the smoothing filter employed. A more intricate exploration of these factors will be conducted in the ablation study section.

We propose potential methods for enhancing the performance of our system concerning both the main agent and interlocutor speech appropriateness. Initially, the model could incorporate interlocutor gestures, audio, and text as conditioning factors. Secondly, incorporating a more extensive history of features from both the main agent and interlocutor into the conditioning process might yield improved gesture generation. Thirdly, the meticulous design of the text embedding model and gesture autoencoder could enhance semantic conditioning and the inherent naturalness of the generated gestures, respectively. These specific aspects will be the focal points of our future works.

**Table 3: Summary statistics of user-study responses from appropriateness for main agent speech, with confidence intervals for the mean appropriateness score(MAS) at the level $\alpha = 0.05$. "Pref. matched" identified how often test-takers preferred matched motion in terms of appropriateness, ignoring ties.**

| Condi-tion | MAS | Pref. matched | Raw response count | | | | | |
|---|---|---|---|---|---|---|---|---|
| | | | 2 | 1 | 0 | −1 | −2 | Sum |
| NA | 0.81±0.06 | 73.6% | 755 | 452 | 185 | 217 | 157 | 1766 |
| SG | 0.39±0.07 | 61.8% | 531 | 486 | 201 | 330 | 259 | 1807 |
| SJ | 0.27±0.06 | 58.4% | 338 | 521 | 391 | 401 | 155 | 1806 |
| BM | 0.20±0.05 | 56.6% | 269 | 559 | 390 | 451 | 139 | 1808 |
| SF | 0.20±0.06 | 55.8% | 397 | 483 | 261 | 421 | 249 | 1811 |
| SK | 0.18±0.06 | 55.6% | 370 | 491 | 283 | 406 | 252 | 1802 |
| SI | 0.16±0.06 | 55.5% | 283 | 547 | 342 | 428 | 202 | 1802 |
| SE | 0.16±0.05 | 54.9% | 221 | 525 | 489 | 453 | 117 | 1805 |
| BD | 0.14±0.06 | 54.8% | 310 | 505 | 357 | 422 | 220 | 1814 |
| SD | 0.14±0.06 | 55.0% | 252 | 561 | 350 | 459 | 175 | 1797 |
| SB | 0.13±0.06 | 55.0% | 320 | 508 | 339 | 386 | 262 | 1815 |
| SA | 0.11±0.06 | 53.6% | 238 | 495 | 438 | 444 | 162 | 1777 |
| SH | 0.09±0.07 | 52.9% | 384 | 438 | 258 | 393 | 325 | 1798 |
| SL | 0.05±0.05 | 51.7% | 200 | 522 | 432 | 491 | 170 | 1815 |
| SC | −0.02±0.04 | 49.1% | 72 | 284 | 1057 | 314 | 76 | 1803 |

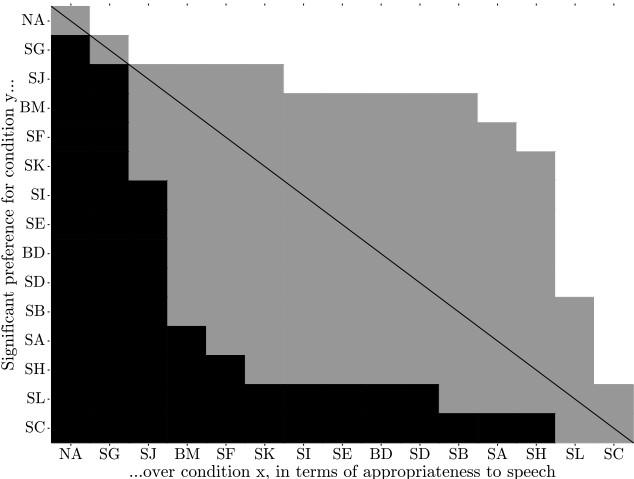

**Figure 5: Significant differences between conditions in the appropriateness for main agent speech. White means the condition listed on the $y$-axis achieved a MAS significantly above the condition on the $x$-axis, black means the opposite ($y$ scored below $x$), and grey means no statistically significant difference at level $\alpha = 0.05$ after correction for the false discovery rate.**

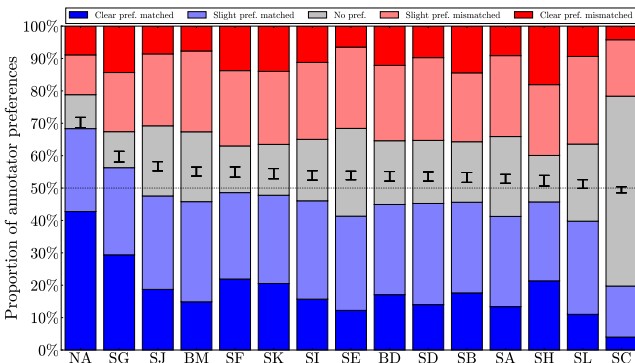

**Figure 4: Bar plots visualizing the response distribution in the appropriateness for main agent speech. The blue bar(bottom) represents responses where subjects preferred the matched motion, the light grey bar(middle) represents tied responses, and the red bar(top) represents responses preferring mismatched motion, with the height of each bar being proportional to the fraction of each category. Lighter colors correspond to slight preference, and darker colors to clear preference. On top of each bar is also a confidence interval for the mean appropriateness score, scaled to fit the current axes. The dotted black line indicates chance-level performance.**

**Table 4: Summary statistics of user-study responses from appropriateness for interlocutor speech, with confidence intervals for the mean appropriateness score(MAS) at the level $\alpha = 0.05$. "Pref. matched" identified how often test-takers preferred matched motion in terms of appropriateness, ignoring ties.**

| Condi-tion | MAS | Pref. matched | Raw response count | | | | | |
|---|---|---|---|---|---|---|---|---|
| | | | 2 | 1 | 0 | −1 | −2 | Sum |
| NA | 0.63±0.08 | 67.9% | 367 | 272 | 98 | 189 | 88 | 1014 |
| SA | 0.09±0.06 | 53.5% | 77 | 243 | 444 | 194 | 55 | 1013 |
| BD | 0.07±0.06 | 53.0% | 74 | 274 | 374 | 229 | 59 | 1010 |
| SB | 0.07±0.08 | 51.8% | 156 | 262 | 206 | 263 | 119 | 1006 |
| SL | 0.07±0.06 | 53.4% | 52 | 267 | 439 | 204 | 47 | 1009 |
| SE | 0.05±0.07 | 51.8% | 89 | 305 | 263 | 284 | 73 | 1014 |
| SF | 0.04±0.06 | 50.9% | 94 | 208 | 419 | 208 | 76 | 1005 |
| SI | 0.04±0.08 | 50.9% | 147 | 269 | 193 | 269 | 129 | 1007 |
| SD | 0.02±0.07 | 52.2% | 85 | 307 | 278 | 241 | 106 | 1017 |
| BM | −0.01±0.06 | 49.9% | 55 | 212 | 470 | 206 | 63 | 1006 |
| SJ | −0.03±0.05 | 49.1% | 31 | 157 | 617 | 168 | 39 | 1012 |
| SC | −0.03±0.05 | 49.1% | 34 | 183 | 541 | 190 | 45 | 993 |
| SK | −0.06±0.09 | 47.4% | 200 | 227 | 111 | 276 | 205 | 1019 |
| SG | −0.09±0.08 | 46.7% | 140 | 252 | 163 | 293 | 167 | 1015 |
| SH | −0.21±0.07 | 44.0% | 55 | 237 | 308 | 270 | 144 | 1014 |

## 5.3    ablation study

We conduct an ablation study to ensure that autoregression is helpful to co-speech gesture synthesis. We calculate Frechet Gesture Distance(FGD), between ground truth and generated motions in the validation set, which are shown in Table 5. As a result, the FGD of discriminator features and raw gestures are improved when using the pre-pose condition.

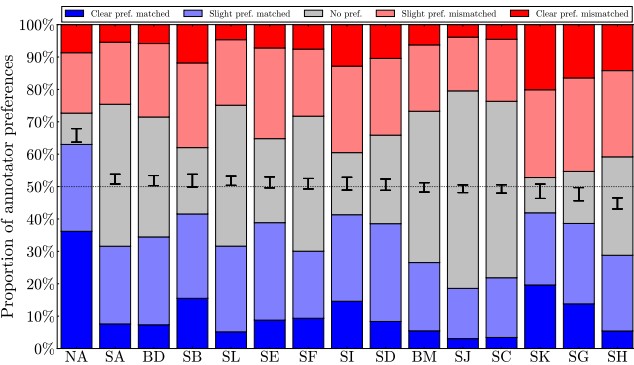

**Figure 6: Bar plots visualizing the response distribution in the appropriateness for interlocutor speech. The blue bar(bottom) represents responses where subjects preferred the matched motion, the light grey bar(middle) represents tied responses, and the red bar(top) represents responses preferring mismatched motion, with the height of each bar being proportional to the fraction of each category. Lighter colors correspond to slight preference, and darker colors to clear preference. On top of each bar is also a confidence interval for the mean appropriateness score, scaled to fit the current axes. The dotted black line indicates chance-level performance.**

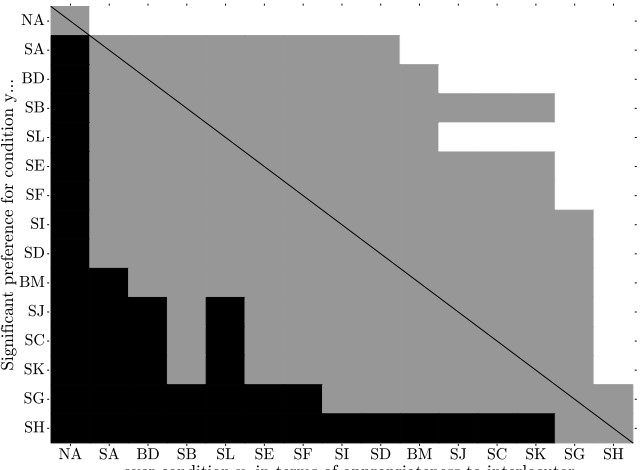

**Figure 7: Significant differences between conditions in the appropriateness for interlocutor speech. White means the condition listed on the $y$-axis achieved a MAS significantly above the condition on the $x$-axis, black means the opposite ($y$ scored below $x$), and grey means no statistically significant difference at level $\alpha = 0.05$ after correction for the false discovery rate.**

## 6 CONCLUSION

In this paper, we introduce an innovative diffusion-based co-speech gesture generation framework that has been submitted to the GE-NEA Challenge 2023. Our approach aims to produce co-speech

**Table 5: Effects of autoregression.**

| Model | FGD(feature) | FGD (raw) |
|---|---|---|
| w/o. pre-pose | 154.984 | 4977.059 |
| w. pre-pose | 77.909 | 2279.612 |

gestures of high fidelity, achieved by proposing a gesture autoencoder for effective domain transfer between the gesture space and latent feature space. Furthermore, we leverage denoising diffusion probabilistic models to address the challenge of co-speech gesture generation. While the comprehensive results indicate that our method achieves a preference-matching score of 54.8% and 53.5% for appropriateness of main agent speech and interlocutor speech, respectively.

Moreover, we conduct an in-depth ablation stud to affirm the utility of autoregressive methods in co-speech gesture synthesis. Our conclusion highlights the strengths of our system in timing synchronization and the generation of contextually fitting gestures for interactive scenarios. Additionally, we propose several forthcoming challenges for research, such as refining the structures of semantic embeddings and gesture embedding models. Our hope is that our approach contributes not only to the advancement of diffusion-based gesture generation research but also finds application across various gesture generation domains.

## ACKNOWLEDGMENTS

This work was supported by the "Development of cognitive/response advancement technology for AI avatar commercialization" project funded by the Brand Engagement Network(BEN)[Q2312881].

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
