# OpenReview forum: "The KU-ISPL entry to the GENEA Challenge 2023-A Diffusion Model for Co-speech Gesture generation"
_ACM.org/ICMI/2023/Workshop/GENEA_Challenge — GENEA Challenge 2023 Workshopproceeding_

### Official Review · Reviewer_W8rh · 2023-07-29
**This entry does not seem to provide any significant contributions.**

**Rating:** 3
**Confidence:** 3

**Review:**

The manuscript was a bit difficult to read due to poor English quality. It is highly recommented that the authors run their future submissions through a proofing service. There are even a few loose incomplete sentences which are totally disconnected. In general there were parts that were difficult to follow due to the poor discourse. Just an example: "To optimize the diffusion model, the variational lower bound on negative log-likelihood."

The approach does not seem to be very original, given that a few other authors have experimented with diffusion-based motion generation and other similar mechanisms described in this approach.
The authors present a decent literature review so they seem knowledgeful of the current state of the art.
The authors do not clarify what, if any part of their contribution, is novel, or and enhancement, compared to existing methods.

The system presented considers only the main agent audio and text manuscript, and fully ignores the interlocutor's information.
It scored quite low compared to other entries, in terms of human-likeness and appropriateness for main agent speech.
By mere chance, it seems to have scored very high on the appropriateness for the interlocutor's speech.
The authors claim this as an achievement which to me, sounds very dishonest.

They affirm that their system fully ignores the data from the interlocutor's speech, therefore, the fact that they scored high on this measure can only mean that the scoring methodology is flawed, and that this happened by mere chance.
This could have been mentioned, however they seem to claim that it was an achievement.

Given that that I did not find a significant novel contribution, any advancement against the current state of the art, namely because the approach scored very low in general (and only by change in the interlocutor case), plus the fact that it is difficult to read and follow, and that it seems to me that the authors were dishonest and overclaimed that their system performs well against the interlocutor's speech when, despite the score, it is clearly impossible given that it ignores such information, I recommend rejecting this entry.

---

### Official Review · Reviewer_J2iu · 2023-07-30
**The paper propose a diffusion model architecture using latent diffusion and autoregressive training for co-speech gesture synthesis. While the evaluation results for human-likeness are not great, it produces the best result in appropriateness for interlocutor speech. Given the design choices and evaluation results, the paper will provide useful insights for the workshop and challenge attendees.**

**Rating:** 7
**Confidence:** 4

**Review:**

## Summary

The paper propose a diffusion-based method for synthesizing co-speech gestures. It utilizes an autoencoder to project the raw gesture motions into the latent space. The diffusion process is trained within the latent space based on the speech and text input conditions. Although the method did not produce very good results in the human-likeness study, it produces the best results in appropriateness for interlocutor speech.

## Strength

The proposed diffusion model utilizes the latent space and is trained in an autoregressive manner. The model is novel and shows good potential to produce high quality gestures. The autoregressive framework also shows to produce better FGD than model trained without pre-poses. While the subjective results in human-likeness are not great, the model performs the best in appropriateness for interlocutor speech. The method is  described in good details with all hyperparameter information for reproducibility. The overall paper exposition is clear and easy to follow.

## Weakness

The following papers that applied latent diffusion model in motion synthesis should also be cited and discussed :

Ao, Tenglong, Zeyi Zhang, and Libin Liu. "GestureDiffuCLIP: Gesture diffusion model with CLIP latents." _arXiv preprint arXiv:2303.14613_ (2023).

Chen, Xin, et al. "Executing your Commands via Motion Diffusion in Latent Space." _Proceedings of the IEEE/CVF Conference on Computer Vision and Pattern Recognition_. 2023.

Some training details could be better discussed. For example, it is not obvious if the latent space autoencoder model is trained in an earlier stage or jointly with the diffusion model. In addition to the data processing hyperparameters, the parameters used in training will also be very helpful for replication.

## Justification of Rating

The paper propose a diffusion model architecture for co-speech gesture synthesis. It utilizes the latent space learning and train the model in an autoregressive manner. The ablation study shows the improvements in FGD with pre-poses to justify the design choice. While the evaluation results for human-likeness are not great, it produces the best result in appropriateness for interlocutor speech. Given a few other concurrent works utilizing diffusion models for motion synthesis, the paper will provide interesting comparison and discussion about the design choices for the workshop and challenge attendees.

---

### Decision · Program_Chairs · 2023-08-04

**Decision:**

Accept (Workshop proceeding)

**Comment:**

The opposite results of the proposed system are quite interesting, and I believe this is worth discussing at the workshop. But there are two main concerns: 1) overclaiming the proposed system for the appropriateness to the interlocutor, 2) poor presentation and lack of details. Overclaiming the system is particularly problematic and would mislead readers as Reviewer W8rh pointed out. The novelty of the proposed method is also limited. The chairs thus decided to accept this paper to the Workshop track to be published in the Adjunct ICMI Proceedings.

Please read the reviews carefully and prepare the camera-ready paper in accordance.
In particular,
* Do not oversell your system. Discussing the mixed results of your system would give a unique contribution to the community.
* Proofread the paper.
* Cite recent relevant papers.
* Add training details.